# Experimental Investigation on the Mass Diffusion Behaviors of Calcium Oxide and Carbon in the Solid-State Synthesis of Calcium Carbide by Microwave Heating

**DOI:** 10.3390/molecules26092568

**Published:** 2021-04-28

**Authors:** Miao Li, Siyuan Chen, Huan Dai, Hong Zhao, Biao Jiang

**Affiliations:** 1Green Chemical Engineering Research Center, Shanghai Advanced Research Institute, Chinese Academy of Sciences, Shanghai 201210, China; limiao@sari.ac.cn (M.L.); daihuan2018@sari.ac.cn (H.D.); 2Green Chemical Engineering Research Center, Shanghai Advanced Research Institute, University of Chinese Academy of Sciences, Beijing 100049, China; 3Shanghai Green Chemical Engineering Research Center, Shanghai Institute of Organic Chemistry, Chinese Academy of Sciences, Shanghai 200032, China; chensy@sioc.ac.cn

**Keywords:** microwave heating, diffusion intensification, calcium oxide, carbon, calcium carbide

## Abstract

Microwave (MW) heating was proven to efficiently solid-synthesize calcium carbide at 1750 °C, which was about 400 °C lower than electric heating. This study focused on the investigation of the diffusion behaviors of graphite and calcium oxide during the solid-state synthesis of calcium carbide by microwave heating and compared them with these heated by the conventional method. The phase compositions and morphologies of CaO and C pellets before and after heating were carefully characterized by inductively coupled plasma spectrograph (ICP), thermo gravimetric (TG) analyses, X-ray diffraction (XRD), scanning electron microscopy (SEM), and X-ray photoelectron spectroscopy (XPS). The experimental results showed that in both thermal fields, Ca and C inter-diffused at a lower temperature, but at a higher temperature, the formed calcium carbide crystals would have a negative effect on Ca diffusion to carbon. The significant enhancement of MW heating on carbon diffusion, thus on the more efficient synthesis of calcium carbide, manifested that MW heating would be a promising way for calcium carbide production, and that a sufficient enough carbon material, instead of CaO, was beneficial for calcium carbide formation in MW reactors.

## 1. Introduction

Calcium carbide (CaC_2_), mainly used to produce acetylene and acetylene-derived products [1,2,3,4,5,6,7,8,9,10], is a vital chemical platform product. The production of CaC_2_ via carbon and calcium oxide to produce acetylene is an essential sustainable chemical process, which recycles calcium and consumes a wide range of carbon, including coal [1,2,3,4,5,6,11] and biomass [12,13]. Nowadays, the acetylene chemical industry has relatively declined, mainly caused by the strong position of the ethylene industry since the 1960s, but in many countries with abundant coal and less oil, coal to carbide is one of the primary processes in the coal chemical industry. In China, the annual production of CaC_2_ was more than 25.88 million tons in 2019 [14].

As we know, calcium carbide production is an energy-intensive process. It is industrially produced in an electric arc furnace at a very high reaction temperature of up to 2000–2200 °C, and at high energy consumption of 3300 kW·h·ton^−1^ standard calcium carbide [3,4,5,15]. Partly due to the energy-intensive nature of the calcium carbide production, academic research and technology improvement on coal to calcium carbide is relatively limited, compared with coal to olefins mainly by Fischer–Tropsch synthesis or MTO processes. In the past decades, some efforts have been carried out to develop new CaC_2_ production technologies to reduce the energy consumption and production cost, such as the rotary kiln-based process [16], the spout-fluid bed reactor with a plasma torch [17,18], and the oxygen-thermal method [4]. However, most of these technologies were abandoned at the pilot scale. The major obstacle may be attributed to the scale-up of reactors. For example, the oxygen-thermal method, the focus of numerous studies in recent years [4], has also suffered the scale-up due to the significant effect of CO pressure on the chemical reaction equilibrium. Liu et al. has studied the oxygen-thermal method with fine feeds and they found that in the small crucible of a thermogravimetric analysis (TGA) system, CaC_2_ was produced at temperatures of ~1750 °C [3,4,5], but in the oxy-thermal CaC_2_ furnace, the appropriate reaction temperature was as high as 2200 °C because of the larger CO partial pressure originating from carbon combustion [4].

In recent years, MW heating has attracted intense attention owing to its advantages of overall, volumetric, instantaneous, and selective heating, high energy efficiency, cleanliness, and safety [19,20,21,22,23]. Especially in the fields of chemistry and chemical industry, MW heating is attractive due to its unique ability to promote mass transfer [24,25,26] and/or accelerate reaction kinetics [27,28,29,30,31,32]. Solid-state processing at high temperatures, such as MW synthesis of carbides, is probably the most enormous potential area for energy savings from MW processing. Recently, MW synthesis of CaC_2_ further evidenced the advantages of MW heating. Pillai et al. [33] reported that 71.8% CaC_2_ was solid-state formed at 1700 °C for 30 min in a 2450 MHz MW reactor, while conventional heat treatment at 1700 °C for 30 min showed 14.1% CaC_2_; in both cases, about 10 g pressed pellets of CaO and graphite fine powder mixtures ball milled in acetone was used. The authors attributed the reaction enhancement and high CaC_2_ yield to the volumetric heating nature of microwaves. 

We have also been carrying out research on the solid-state synthesis of calcium carbide by MW heating in these years [34]. We validated the greater efficiency and productivity of the MW synthesis of calcium carbide and carried out the kilogram scale pilot using a 915 MHz microwave furnace (shown in Figure 1). The acetylene gas output of produced calcium carbide reached 300 mL/g after reacting at 1720 °C for 1 h when the molar ratio of semi-coke to CaO was 3, suggesting an optimistic prospect of the scale-up of the MW reactor. Moreover, the mass-production of 915 MHz microwave sources with a power of 100 kilowatts has provided technical support to the promising alternative route for CaC_2_ industrial production, too. However, detailed information on the mass transmission associated with MW heating has not been clear, which would hamper the further development of the MW reactor. We know efficient enough diffusion of reactants is usually the prerequisite for multiphase solid-state reactions, and mass transfer is even the control-step in some cases. Binner et al. found that the diffusion was the rate-controlling step in the MW synthesis of titanium carbide [30].

There is some literature on the diffusion behaviors in the CaC_2_ production process by conventional heating. However, to the best of our knowledge, no unified view on it has been achieved. In earlier research, Kameyama [35] and Tagawa [36] proposed that the solid-state synthesis of CaC_2_ was based on the interdiffusion of C and CaO. Muller [37,38,39] later proposed that carbon ironically transported into the CaO lattice and enables the C_3_ molecules to form another “interstate” compound (CaC_3_O). El-Naas [18] studied the solid-phase synthesis of CaC_2_ in a plasma reactor and suggested that carbon diffused into the calcium oxide. Recently, Li et al. [12,13,40] studied the CaC_2_ formation process using high-temperature TG-MAS and reported that CaO diffused to C in the solid phase reaction stage. As for MW synthesis of CaC_2_, Pillai et al. [34] did not pay much attention to this point, and they suggested that MW heating promoted the formation of CaC_2_-CaO eutectics the same as in the conventional field, which accelerated the diffusion of CaO, hence forming more CaC_2_ in the MW field. In fact, due to the unique selective heating nature of MW, the reactants coupling well with MW irradiations would exhibit different diffusion behaviors from those by conventional heating, therefore causing different kinetic behaviors and operation conditions in MW reactors. We know that carbon acts as a combined reactant and susceptor via an ohmic dissipation mechanism in this system. Hence, systematic studies to reveal the diffusion behaviors of calcium oxide and carbon in the MW field are crucial for MW reactor design and process optimization.

In this study, the diffusion behaviors of C and CaO were experimentally investigated in the MW field and compared with those in the conventional field. A suite of diagnostic techniques including ICP, TG, XRD, XPS, and SEM-EDS were used to monitor the morphology, composition, and chemical state of graphite and calcium oxide pellets before and after heating. After that, the influence of microwaves on the diffusion process was elucidated.

## 2. Materials and Methods

### 2.1. Materials

Analytical-grade calcium oxide (CaO > 99.8%, from Greagent) was calcined at 1000 °C for 6 h followed by crushing and grinding to obtain CaO with a grain size of about 500 mesh. Analytical-grade graphite (graphite, >99.99%, from Greagent, particle size was about 200 mesh) was treated at 300 °C for 6 h under N_2_ flow. About 8 g treated CaO powder or graphite powder was carefully pressed into pellets (40.5 mm in diameter) at 60 MPa and precisely weighed. In this way, we could assume that the pellets were thick enough for the diffusion while the diffusion elements were concentrated in a thin layer. 

### 2.2. Diffusion Experiments

The diffusion experiments of graphite and CaO were carried out in a multi-mode high-temperature vacuum microwave oven with a maximum output power of 4.5 kW at 2.45 GHz (Changsha Longtai Microwave Thermal Engineering Co., Ltd., Changsha, China), as shown in Figure 1. In order to obtain a convincing measurement on temperature, an infrared thermometer was used to detect the temperature of materials. Simultaneously, three temperature measuring rings wrapped by graphite paper were respectively placed at the upper, middle, and bottom of the heated material to correct the infrared temperature measurement. The comparative tests were carried out in a corundum tube furnace (Shanghai Chenhua Electric Furnace Co., Ltd., Shanghai, China, the maximum service temperature was 1650 °C). In diffusion experiments, four pieces of graphite pellets and three pieces of calcium oxide pellets with smooth surfaces were interlaced like a burger for heating, as shown in Figure 2. Noticeably, in the MW field, the burger was placed in a boron nitride crucible, which was infilled with a layer of 1 cm thick graphite powders beforehand to avoid spark generation. The crucible containing samples was placed in the corundum tube or in the microwave transparent insulation barrel (an infrared temperature measuring hole in the center of its lid). In order to improve the uniformity of microwave irradiation, the storage chassis, on which the microwave transparent insulation barrel was put, rotated at a speed of 20 revolutions per minute. The samples were heated to the target temperature (1320–1620 °C) in 2 h and held for 1 h by adjusting the MW output power under the flow of Ar (80 mL.min^−1^). The relatively low heating rate in the MW field would diminish the possible temperature heterogeneity between CaO and graphite pellets, which was possibly caused by their different microwave absorbing properties. As for electric heating, considering the poor resistance of the corundum tube to rapid cooling and heating, the temperature of the electric tube furnace was firstly raised to 1000 °C, then the crucible containing samples was carefully put in and heated to the target temperature at a rate of 2 °C.min^−1^. Thus, the whole heating up time by electric heating was much longer than that by MW heating, meaning a longer diffusion time in the conventional thermal field. After cooling to 300 °C, the treated samples were quickly transferred to a glove box for XRD, TG, and ICP test preparation, and the remaining samples were sealed and stored in a dry box filled with nitrogen for other analyses. 

### 2.3. Sample Analysis

For TG, XRD, and microstructure observation analysis by SEM, sample powders were prepared by cutting a quarter of a target pellet and then grounding evenly in an agate mortar operated in a glove box. For XPS analysis, the sample powders were carefully scraped down from the contacted surface of another quarter of the target sample. As for the contacted surfaces analysis by SEM and EDS, another quarter of the target sample was used for analysis directly. The diffused Ca amount was examined by ICP on Thermo iCAP7600, while the diffused C amount was determined by TG analysis performed on a TA SDT Q600 in He/O_2_ flow, in which the value was the weight loss in O_2_ flow subtracted by the weight loss in He flow so that the possible interference of calcium carbonate could be avoided. XRD was recorded with a Rigaku MiniFlex 600 X-ray diffractometer, using filtered Cu Kα (λ = 0.15406 nm) as the radiation source operating at 40 kV and 15 mA and operating with the step size of 0.02°. SEM and EDS mapping were observed using a JEOL JSM-7800F Prime, and XPS was carried out on a Kratos Axis Ultra DLD operating at 30 eV. The dielectric permittivity test used a 2018D1F5 network analyzer for measurement.

## 3. Results and Discussion

### 3.1. Morphology Observations

In order to investigate the changes of the phase composition and texture morphologies before and after MW heating, we carried out the characterization of the samples. Figure 3a shows the photographs of graphite and CaO pellets before and after the typical MW heating process at 1520 °C. Clearly, the heated CaO pellet turned from white to gray, while the heated graphite pellet kept the dark black color. Beyond that, the heated CaO pellet exhibited an obvious contracted outsize from 40.5 × 6.3 mm to 34.5 × 4.6 mm, but the heated graphite pellet expanded from 40.5 × 3.8 mm to 42.0 × 4.3 mm. This was because of their different thermal expansion properties for graphite and calcium oxide. Figure 3b–e illustrates the SEM images of graphite and CaO particles in the heated and unheated pellets. The results demonstrated the remarkable sintering and agglutination of CaO particles, while no pronounced morphology change was observed for graphite particles.

Figure 4 further exhibits the SEM images of their contacted surfaces after MW heating at 1520 °C, on which elemental mapping analysis was induced. Clearly, in Figure 4a the more compact microstructure was observed on the CaO pellet surface, compared with that on the graphite pellet surface shown in Figure 4a′, suggesting the concentration and aggregation of CaO particles after MW heating. Figure 4b,c showed the corresponding EDS mapping diagrams on the heated CaO pellet surface, suggesting the distributions of Ca and C on the CaO pellet surface. The results revealed that a large amount of C was uniformly distributed on the contacted surface of calcium oxide pellets, indicating that C easily diffused to the calcium oxide pellet under microwave heating. The EDS mapping diagram in Figure 4b′,c′ showed C and Ca distribution on the contacted surface of the graphite pellet. The larger spots in Figure 4c′ should be the calcium oxide particles which adhered on the graphite pellet surface. The compositional distribution suggested a smaller diffused amount of Ca atoms into the graphite pellet. The mapping results suggested that CaO and C inter-diffused, but C diffused faster.

### 3.2. Phase Structure Analysis

XRD analysis was performed to investigate the MW heating temperature on the phase composition of the heated pellets. The obtained results are shown in Figure 5. For the unheated calcium oxide sample shown in Figure 5a, the diffraction peaks at 32.5°, 37.7°, 54.1°, 64.4°, and 67.6° could be noted, which corresponded to (111), (200), (220), (311), and (222) of calcium oxide, respectively. After MW heating, a new peak of graphite (002) at 26.6° appeared, and its intensity increased with an increase in the temperature. The results confirmed the diffusion of C to calcium oxide, and the diffused amount increasing with an increase in temperature. Meanwhile, the diffraction peak intensities of CaO also increased with the increasing temperature, suggesting an increasing grain size, which coincided with the microstructure showed by SEM.

Figure 5b exhibited the XRD patterns of the corresponding graphite pellets. It can be seen that, except for the diffraction peaks at 26.5° and 54.5° attributing to (002) and (004) diffraction peaks of graphite, respectively, there was no other apparent characteristic diffraction peaks to calcium compounds, such as CaO and CaC_2_. This should be because the peak strength of graphite was much higher than that of calcium oxide, which weakened the signals of calcium oxide. As shown in Figure 5b, the mixture of calcium oxide and graphite with a molar ratio of 1:1 exhibited very strong graphite characteristic peaks but extremely weak CaO signals.

### 3.3. Chemical State Analysis

In order to further explore the chemical states of Ca and C on the contacted surfaces of MW heated samples, the microwave heated samples heated at 1470 °C were investigated by XPS analysis. Figure 6a shows the survey spectrum of their contacted surfaces. Both the calcium oxide and graphite pellets contained the three elements of C, O, and Ca. Their molar percentages of C, O, and Ca on the surface of the calcium oxide plate were 40.2%, 44.1%, and 15.7%, respectively, while on the corresponding graphite plate surface molar percentages of them were 89.1%, 8.2%, and 2.7%, respectively. The above result further indicated the faster diffusion of C in the MW heating field. 

Figure 6b–d showed the high-resolution Ca 2p, O 1s, and C 1s spectra on the contact faces of the calcium oxide and graphite pellets. As can be seen from Figure 6b, both the graphite and calcium oxide pellets showed two characteristic peaks. The peak at 347.3 eV was ascribed to Ca 2p_3/2_ of CaCO_3_, and the other one at 350.8 eV was ascribed to Ca 2p_1/2_ of CaCO_3_, suggesting calcium compounds changed to CaCO_3_ in the XPS analysis process [34]. Two small but obvious signals of Ca 2p_3/2_ and Ca 2p_1/2_ over the graphite pellet were observed, manifesting Ca diffusion from the CaO pellet to the graphite pellet. Figure 6c exhibited O1s signals for both calcium oxide and graphite pellets. The calcium oxide pellet gave a high-intensity peak near 531.9 eV, which was the characteristic peak of O1s for CaCO_3_, but the graphite pellet offered a much weaker peak here. The results in Figure 6b,c indicated that a small amount of Ca diffused to the graphite pellet, which also agreed with the described SEM and XRD results. Figure 6d displayed C 1s signals on the contacted surfaces of the CaO and graphite pellet. The high-strength peak at 284.5 eV was the characteristic peak of graphite carbon, and the weak peak near 289.8 eV corresponded to C1s of CaCO_3_. Clearly, the graphite pellet showed the very strong C1s signal for graphite carbon at 284.5 eV and a hardly visible C1s signal for CaCO_3_ at 289.8 eV, but the CaO pellet exhibited a middle-intensity peak at 284.5 eV and a weaker signal at 289.8 eV. The result further suggested that a large amount of graphite C diffused from the graphite pellet to CaO pellet, but a smaller amount of Ca diffused from the CaO pellet to the graphite pellet. 

### 3.4. The Diffusion Amount of C and Ca

In order to know better about the diffusion behaviors of C and CaO by MW heating, we quantitatively analyzed the diffused C amount and Ca amount at different heating temperatures and compared them with those by electric heating. Figure 7a showed the results by electric heating. C was checked out in the heated CaO pellet, and Ca was also checked out in the heated carbon pellet, suggesting that C and Ca inter-diffused by electric heating. In detail, the diffusion amount of C and Ca increased with temperature increasing at lower temperature. When the temperature was higher than 1520 °C, the diffused Ca amount unexpectedly decreased and showed a distinct volcano shape with the increasing temperature, while the diffused C amount continued to increase slowly, and became higher than the corresponding Ca amounts. According to $D=D_{0}\times e^{\frac{-Q}{\mathrm{RT}}}$, the diffusion rate should increase exponentially with the increase of temperature. The results suggested a different kinetic behavior at a higher heating temperature. The diffusion of C and Ca, especially Ca, were hindered at a higher temperature, causing the diffused C amount to become relatively greater than that of Ca. The results agreed with the report that C and CaO inter-diffused by Yamanaka [35] and Tagawa [36], but far away from the proposal that CaO should diffuse to C reported by Li et al. [12,13]. The great influence of temperature on the diffusion behaviors may be partially caused by these different reports.

Figure 7b showed the effect of heating temperature on the diffusion amounts of C and Ca by MW heating. Clearly, the carbon amount in heated CaO pellets constantly increased with temperature, and the obstruction to C diffusion in a high temperature range became indistinct. Moreover, the carbon amount in the heated CaO pellet was much higher than that heated by the traditional electric heating. In our experiments, the whole heating time in the MW field was much shorter than that by traditional heating, so that the significantly improved carbon diffusion amount indicated the intensification of MW irradiation on C diffusion.

As for the diffused Ca amount, a more obvious volcano shape was observed over the MW heated samples. The calcium amount at first increased with temperature, and was clearly higher than those by electric heating but lower than the corresponding C amount. That is to say, by MW heating, Ca diffusion was slower than the corresponding C diffusion, but faster than that heated by the electric method. It indicated that MW heating showed a positive effect on Ca diffusion, but the intensification effect was not as great as that on graphite. Moreover, by MW heating the maximum diffusion amount of Ca appeared at 1420 °C, which was 100 °C lower than that by traditional heating. Then, the Ca diffusion amount quickly decreased with the increase of temperature until 1570 °C. The result suggested that Ca diffusion to C was blocked at a lower temperature by MW irradiation. Additionally, by continuing to increase the heating temperature over 1570 °C, the Ca diffusion amount increased but was still maintained at a low level. The results indicated that the obstruction effect on Ca diffusion did not disappear or get weaker at a higher temperature and ran through the whole high temperature range.

From the above experimental results, it was clear that there was a “microwave effect” on the mass transfer of calcium oxide and carbon in the solid-state synthesis of calcium carbide. MW irradiation enhanced the diffusion of C and Ca, especially C diffusion to Ca. Whittaker [26] investigated the effects of the electric field produced by MW heating upon the mass transport during the sintering of ceramic materials. He found that the intense electric field MW irradiation produced at the lattice defects and interparticle boundaries would enhance ion mobility, and that the mass transport rate at a given temperature would be enhanced by the presence of MW. This “nonthermal effect” of MW irradiations, resulting from the ponderomotive driving force of the electric field, was a function of both the sample permittivity and the strengths of the high-frequency fluctuating electric field. As shown in Figure 8, the dielectronic constant of graphite at 2450 MHz was more than 20, and its tanδ was over 0.15 in all temperature ranges, suggesting that graphite was an excellent MW absorber. The excellent coupling of graphite with MW enhanced its diffusion to CaO and changed the diffusion trend at high temperatures exhibited in the traditional thermal field. As for CaO, its dielectric constant was about 5.0 and tanδ was about 0.08, much lower than graphite, suggesting its weaker coupling with MW and then weaker MW enhancement on its diffusion behavior.

It should be stated that, MW heating enhancement on the mass transport here, which was obviously not originated from the “thermal effect” of MW irradiations, would have a great benefit to the solid synthesis of CaC_2_. This conclusion was different from the proposal by Pillai et al. [33], that the fast heating rate and higher temperature than those actually observed by MW heating promoted the diffusion of CaO to carbon, hence forming more CaC_2_.

### 3.5. Verification of the Impediment by CaC_2_

On the other hand, it was necessary for us to explore the resistance effect on the diffusions occurring at a high temperature range. In our tested system, there were three chemical elements including C, Ca, and O, so there should be some new chemical material produced in the heating process that hindered the diffusion of Ca. It was reported that the critical temperature for calcium carbide formation could be as low as about 1460 °C [3,40] and 1440 °C [41]. Moreover, Ca^2+^ with a larger ion radius was reported to possess high barrier energy in the calcium carbide crystal [42]. In this study, the inflection point of the Ca diffusion amount was 1470 °C for the MW heating and 1570 °C for the electrical heating. Thus, it might be reasonable for us to propose that CaC_2_ crystals would be formed at the interfaces in both thermal fields, which would obstruct the diffusion of C and Ca, especially the diffusion of Ca.

The thermodynamic analysis proposed that the higher CO partial pressure would inhibit the formation of calcium carbide. Under the CO atmosphere, the critical formation temperature of calcium carbide was above 1780 °C [41]. Thus, the diffusion amount of both calcium and carbon would increase steadily with the increase of temperature under a CO atmosphere in the investigated temperature range. Therefore, in order to confirm the formation of CaC_2_ in the heating processes, we carried out a verification test, in which 250 g of mixed raw materials was MW heated at 1570 °C for 2 h to synthesize CaC_2_. In detail, 560 g of CaO, 360 g of graphite, and 300 g of water were mixed together. The well mixed slurry was then kneaded into spherical balls with a diameter of about 1 cm. After being dried at 120 °C for 24 h, the obtained balls were used as a reactant. In the experiment, the system pressure was kept at about 1000 kPa using a vacuum pump to imitate the low CO partial pressure in the diffusion test. After reaction, the products were analyzed by XRD. Hydrolysis testing was also performed to quantify the synthesized CaC_2_ in the C_2_H_2_ gas generation and collection apparatus.

Figure 9 demonstrated the XRD pattern of the samples. We know four temperature- induced CaC_2_ modifications were experimentally reported, including the common form of CaC_2_-I (*I4/mmm*, *Z = 2*, 139), two metastable low-temperature monoclinic modification CaC_2_-Ⅱ (*C2/c*, *Z = 4*, 15), and CaC_2_-Ⅲ (*C2/m*, *Z = 4*, 12), and cubic high-temperature modification CaC_2_-Ⅳ (*Fm-3m*, *Z = 4*, 225) [43]. From Figure 9 we can see, besides the characteristic peaks of CaO and graphite, there were two small but distinct new peaks at 30.4° and 31.1°, which respectively corresponded to (210) and (004) diffraction peaks of CaC_2_-Ⅲ. Here, no CaC_2_-I was observed, manifesting a low generation temperature of calcium oxide by MW heating. The further hydrolysis testing showed the product with 91 mL/g C_2_H_2_ gas generation, equaling 26 wt % of CaC_2_ in the product, confirming that CaC_2_ was formed at a lower heating temperature.

Figure 10 exhibits the diffusion amounts of Ca and C heated under a CO atmosphere at different temperatures. Unsurprisingly, under a CO atmosphere there was no inflection point observed, and the diffusion amounts of Ca and C constantly increased with the heating temperature. When the temperature was over 1520 °C, the Ca diffusion amount was lower than the C diffusion amount. The results convinced the suppression of the formed calcium carbides on the diffusion of reactants, especially on the diffusion of Ca. The lower inflection temperature of calcium diffusion in the MW field indicated a lower generation temperature of CaC_2_, suggesting that MW heating only accelerated the mass diffusion but also enhanced the chemical reaction.

### 3.6. The Schematic Diagram of the Diffusion Model in the MW Field

Based on our experimental results, it was clear that the diffusion behavior of carbon and calcium oxide should be divided into two different stages within the examined temperature range in the MW field. The diffusion schematic diagram was shown in Figure 11. At a lower temperature, carbon and calcium would directly inter-diffuse on the interfaces, and the diffusion rate increased with temperature. When the temperature was higher, carbon and calcium would partly diffuse through the produced calcium carbide crystals. However, because of better microwave absorption and less atomic radius of carbon, C diffusion in the MW field consistently increased with the increasing temperature and was much faster than that of Ca. Thus, at a lower temperature, the interdiffusion of C and Ca occurred, but at higher temperatures, the diffusion of carbon was prominent in the MW field. In the industrial production of CaC_2_ using an arc-furnace, excess CaO is necessarily used to form CaC_2_-CaO liquid eutectics, which promote CaO diffusion to carbon to accelerate CaC_2_ generation. However, when C was insufficient in the reaction system, the produced CaC_2_ was easy to decompose into Ca and C.
(CaC_2_ + 2CaO = 3Ca + 2CO). 

Thus, excess CaO would accelerate the decomposition of CaC_2_, add the heat load, and reduce carbide content of the product.

## 4. Conclusions

In this study, the diffusion behaviors of C and CaO were experimentally investigated in the MW field and compared with those in the conventional field. The obtained experimental results manifested the high-efficient and dominant diffusion of C toward CaO in the solid-synthesis of CaC_2_ by MW heating. The effective mass transfer of reactants in the MW thermal field was no longer dependent on CaO diffusion to carbon, and excess CaO was not needed to promote the formation of CaO-CaC_2_ eutectics. In essence, carbon reactant played a key role in both mass and heat transfer in the MW reactor. An appropriate C/CaO ratio in the MW reactor was close to the stoichiometric ratio 3, suggesting that in MW reactors, higher purity calcium carbide would be more easily and efficiently obtained.

## Figures and Tables

**Figure 1 molecules-26-02568-f001:**
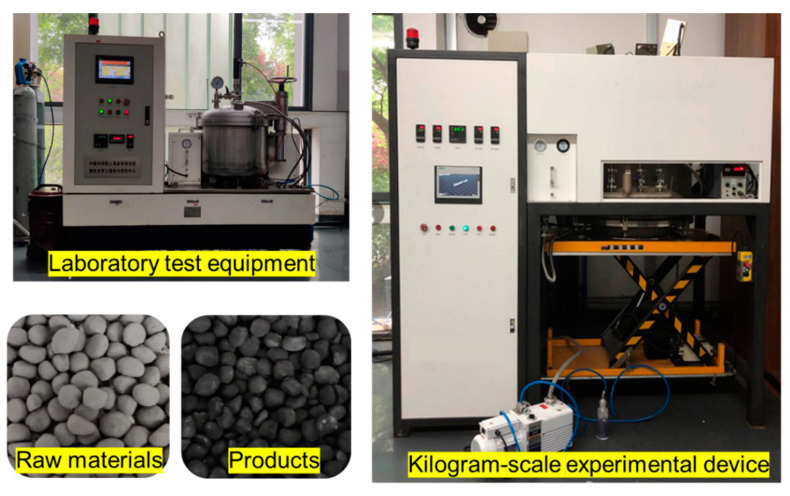
The MW heating equipment for CaC_2_ synthesis.

**Figure 2 molecules-26-02568-f002:**
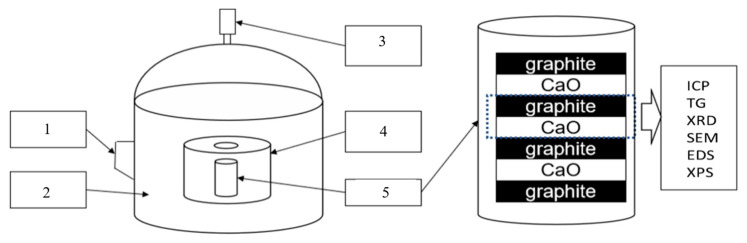
Schematic set-up for microwave heating. 1: microwave source, 2: microwave cavity, 3: infrared thermometer, 4: insulation barrel, 5: crucible.

**Figure 3 molecules-26-02568-f003:**
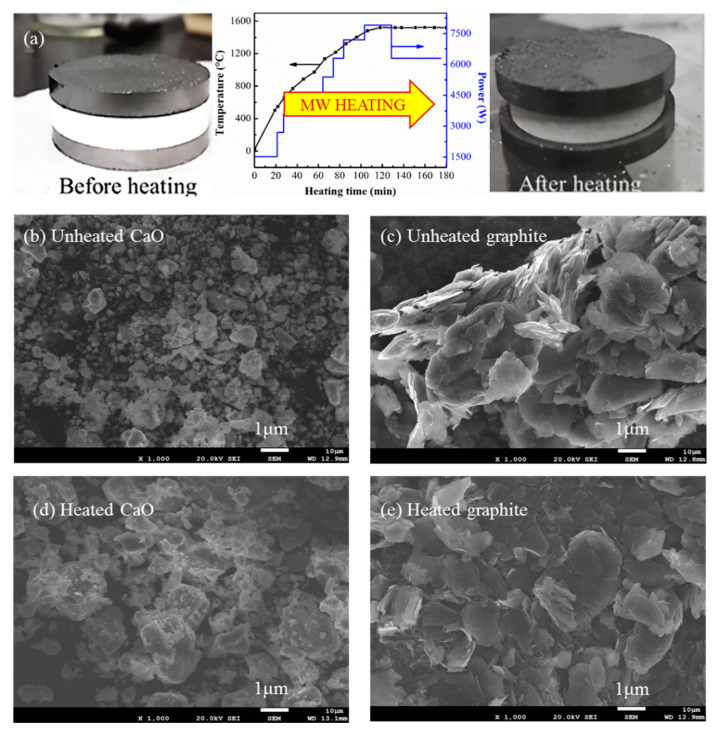
The appearance of CaO and graphite tables before and after MW heating (**a**) and the SEM images of CaO (**b**,**d**) and graphite (**c**,**e**); before (**b**,**c**) and after (**d**,**e**) MW heating at 1520 °C.

**Figure 4 molecules-26-02568-f004:**
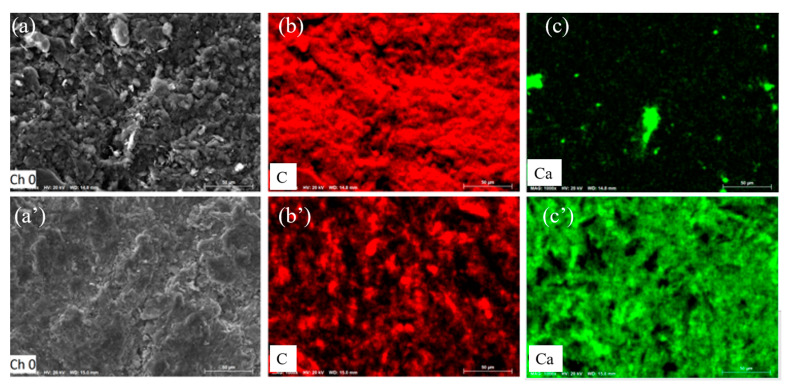
SEM images of the contacted surface of MW heated CaO pellet (**a**) and graphite pellet (**a’**), and the corresponding EDS mapping images of C (**b**,**b′**) and Ca (**c**,**c′**) at 1520 °C.

**Figure 5 molecules-26-02568-f005:**
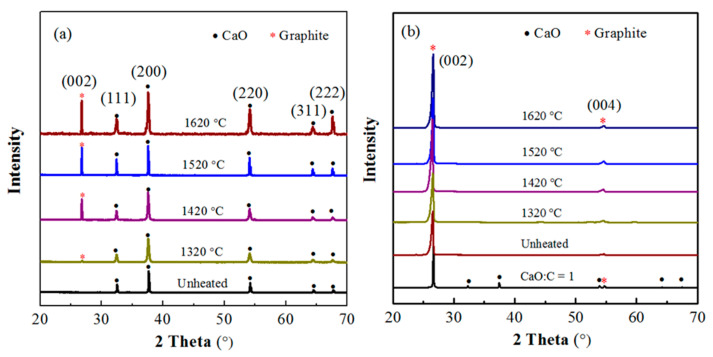
XRD pattern of CaO and graphite samples subjected to microwave treatment in a temperature range of 1320–1620 °C: (**a**) CaO; (**b**) graphite.

**Figure 6 molecules-26-02568-f006:**
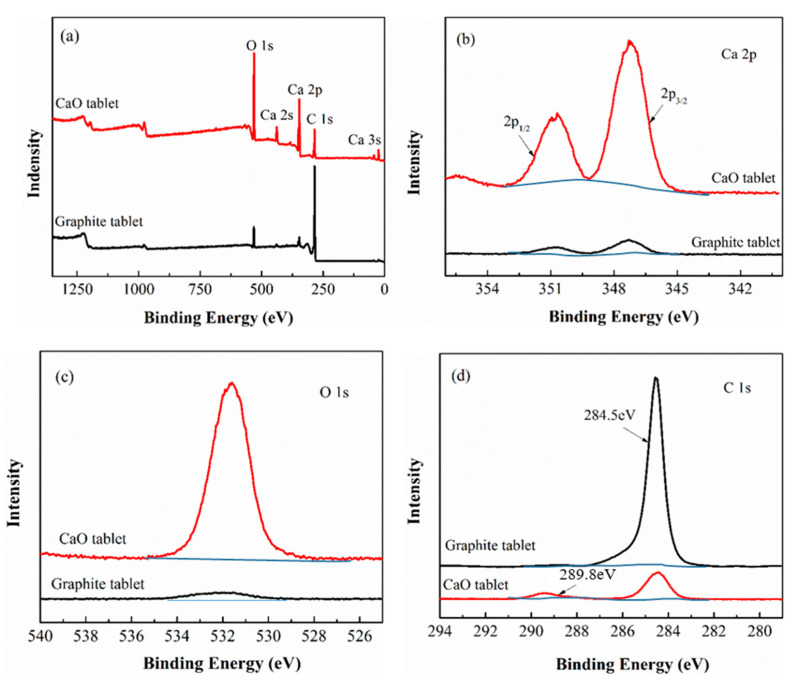
XPS spectra of the microwave-heated CaO and graphite pellets at 1470 °C. (**a**) Survey spectrum, (**b**) high-resolution Ca 2p spectrum, (**c**) high-resolution O 1s spectrum, (**d**) high-resolution C 1s spectrum.

**Figure 7 molecules-26-02568-f007:**
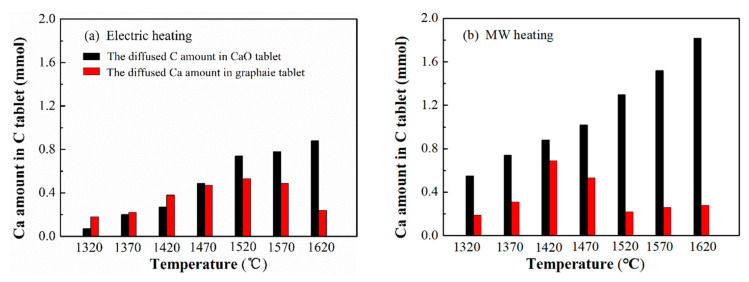
The diffused C amounts in CaO pellets and the diffused CaO amounts in graphite pellets heated by electric heating (**a**) and MW heating (**b**) under Ar atmosphere.

**Figure 8 molecules-26-02568-f008:**
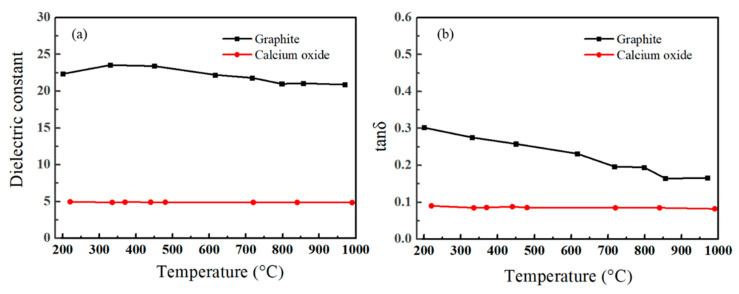
The dielectric constant (**a**) and tanδ (**b**) of graphite and calcium oxide.

**Figure 9 molecules-26-02568-f009:**
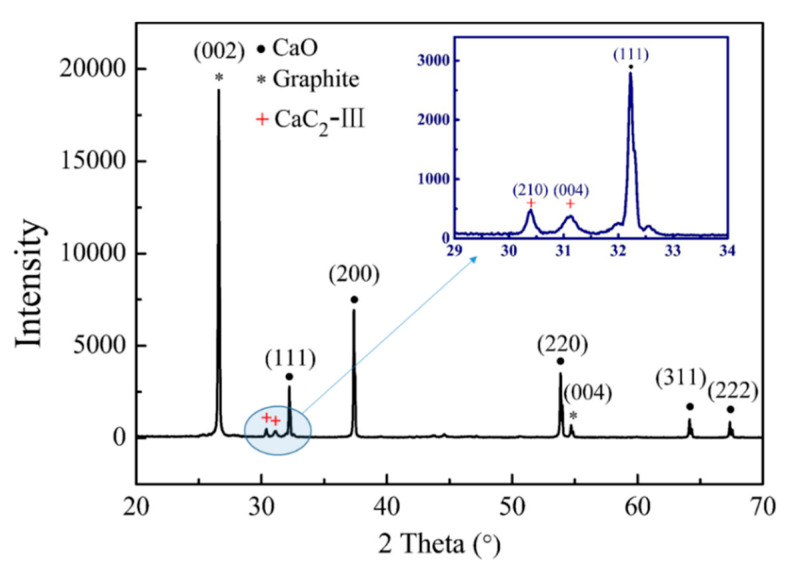
XRD pattern of MW heated product at 1570 °C.

**Figure 10 molecules-26-02568-f010:**
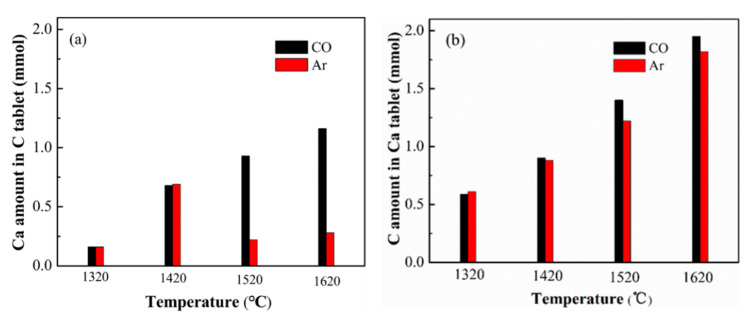
The diffusion amounts of Ca (**a**) and C (**b**) under Ar and CO atmosphere.

**Figure 11 molecules-26-02568-f011:**
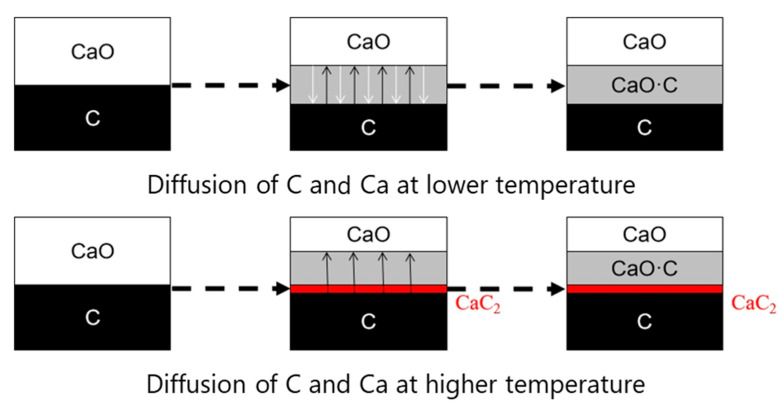
The schematic diagram of the diffusion of C and CaO in the MW field.

## Data Availability

Not available.

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
