# Peer review of "Experimental Investigation on the Mass Diffusion Behaviors of Calcium Oxide and Carbon in the Solid-State Synthesis of Calcium Carbide by Microwave Heating"

_molecules, 2021, doi:10.3390/molecules26092568_

Round 1
Reviewer 1 Report
The article is dedicated to an important theme and will be interesting for wide circle of readers. The introduction gives an adequate description of field of investigation, the results are clearly presented. The reviewer has found several insignificant drawbacks, which should be improved before the publishing.
- There is no description of dielectric permittivity and losses measurement technique. It should be added to Experiment section.
- In line 180 it should be "e" instead of "d" for graphite.
- In lines 90 and 91 in C3 and CaC30 the lower index is missed, I believe.
- In line 410 the sentence "This work was supported by." is unclear.
Reviewer 2 Report
Dear Authors,
I find excellent work donefrom the point of view of analytical chemistry, materials science, and mass transfer phenomena.
The data from the experimental measurements well support the discussions commented on by the authors.
However, there are some suggestions that I would like to make to improve this paper for publication.
These suggestions are as follows:
line 171: The text in the Fig 3a will be: ...ure 3 (a) showed the photographs of graphite and CaO pellets "before and" after... (as it is indicated in the Fig. 3a)
line 182: The text in the Fig 3b to 3e is confusing.
Figure 6d: The value of 289.2 eV should be 289.8 eV, based on the text on lines 245 to 248.
lines 256, 258, 277, 280, 286, 301 (twice), and ...: Please change the word "field" in "thermal field" and "electric field" to "heating" because the term of " electric field "has its own meaning in electromagnetism.
lines 387 to 392: the text from "In the ..." to the word "product." must be out of conclusion.
line 431: Replace "9Wu," with "Wu,"
line 440: Replace "Dc" with "d.c."
In addition, I would like to express that I have problem with some references:
I can not be able to access to nº 14, 37, 38, and 39.
I think that the references nº 34, 41 42 are only in Chinese.
Reviewer 3 Report
Manuscript is interesting because it deals with an energy issue in the area of industrial chemistry. It is well crafted and well specified. It presents some details that need to be improved.
Abstract needs to be reduced, it is too extensive. Accuracy, Brevity and clarity.
Picture 2 needs to specify each one of the elements of the different parts of equipment. Picture 4 needs to show the different components, Ca, C, etc .
In diffusion amount of C and Ca part the authors talk about “volcano shape”, could you explain with more accuracy, what it means.
Authors need to explain why use a reference from 1896 “Morehead, J. T.; de Chalmot, G., THE MANUFACTURE OF CALCIUM CARBIDE. Journal of the American Chemical Society 416 1896, 18, (4), 311-331.”
Round 2
Reviewer 3 Report
Comments and suggestions have been considered